# Species-Level Gut Microbiota Analysis after Antibiotic-Induced Dysbiosis in Horses

**DOI:** 10.3390/ani11102859

**Published:** 2021-09-30

**Authors:** Rebecca Di Pietro, Luis G. Arroyo, Mathilde Leclere, Marcio Carvalho Costa

**Affiliations:** 1Department of Biomedical Sciences, Faculté de Médecine Vétérinaire, Université de Montréal, Saint-Hyacinthe, QC J2S 2M2, Canada; rebecca.di.pietro@umontreal.ca; 2Department of Clinical Studies, Ontario Veterinary College, University of Guelph, Guelph, ON N1G 2W1, Canada; larroyo@uoguelph.ca; 3Department of Clinical Sciences, Faculté de Médecine Vétérinaire, Université de Montréal, Saint-Hyacinthe, QC J2S 2M2, Canada; mathilde.leclere@umontreal.ca

**Keywords:** equine intestinal microbiota, microbiome, PacBio, long-read sequencing

## Abstract

**Simple Summary:**

DNA sequencing has been used to characterize the intestinal microbiota of horses. However, all published studies are based on Illumina technology to date because it is widely accessible, but a major limitation is the short bacterial DNA fragment sequenced. Other long-read sequencing technologies (i.e., PacBio) can classify bacteria at the species level and therefore are more meaningful for the identification of specific changes in the gut after the use of antimicrobials, for example. The objectives of this study were to characterize the gut microbiota of horses at the species level before and after trimethoprim sulfadiazine administration and to compare results with short-read Illumina sequencing. PacBio sequencing failed to classify the equine intestinal microbiota at the species level but detected greater richness and less unclassified bacteria compared to Illumina sequencing. Bacteroidetes were the most abundant bacteria in the feces, followed by Firmicutes and Fibrobacteres in this study. Further studies should focus on improving databanks for equine studies.

**Abstract:**

All current studies have used Illumina short-read sequencing to characterize the equine intestinal microbiota. Long-read sequencing can classify bacteria at the species level. The objectives of this study were to characterize the gut microbiota of horses at the species level before and after trimethoprim sulfadiazine (TMS) administration and to compare results with Illumina sequencing. Nine horses received TMS (30 mg/kg) orally for 5 days twice a day to induce dysbiosis. Illumina sequencing of the V4 region or full-length PacBio sequencing of the 16S rRNA gene was performed in fecal samples collected before and after antibiotic administration. The relative abundance and alpha diversity were compared between the two technologies. PacBio failed to classify the equine intestinal microbiota at the species level but confirmed Bacteroidetes as the most abundant bacteria in the feces of the studied horses, followed by Firmicutes and Fibrobacteres. An unknown species of the Bacteroidales order was highly abundant (13%) and deserves further investigation. In conclusion, PacBio was not suitable to classify the equine microbiota species but detected greater richness and less unclassified bacteria. Further efforts in improving current databanks to be used in equine studies are necessary.

## 1. Introduction

The gut microbiota, a collection of microorganisms mostly dominated by bacteria but also includes viruses, archaea, and fungi, plays a crucial role in immune development [1], energy metabolism [2], and barrier function against pathogens [3]. In horses and other herbivorous species, the gut microbiota is involved in producing short-chain fatty acids, which in turn provide them with energy [4]. A marked imbalance of the healthy gut microbiota is more commonly referred to as dysbiosis and has been reported mainly in horses with colitis [5,6], colic [7], and after antibiotic administration [8,9].

As in other species, antibiotic treatment can induce marked changes in the equine gut microbiota, but the consequences of this are just starting to be unraveled [10]. In addition to the changes in the fecal microbiota composition, treatment with oral antimicrobials (i.e., metronidazole) induced a decrease in diversity [11]. Another study in which healthy horses were treated with either procaine penicillin, ceftiofur sodium, or trimethoprim sulfadiazine (TMS) demonstrated a drug-specific effect on the selection of bacterial populations, with oral TMS having the most profound impact on the microbiota [8].

While culture-based methods have been widely used to detect specific bacteria, many species are challenging to culture using traditional methods and culture media [12]. The development of culture-independent methods allowed for a more complete characterization of the gut microbiota, although major limitations still exist [13]. For instance, while short-read sequencing (e.g., Illumina) can sequence thousands of samples simultaneously in a reasonable time frame, challenges arise when classifying organisms at the species level due to the limited size of the DNA fragments [14]. Illumina platforms can sequence approximately 300 base pairs (bps) per read (depending on the platform chosen), while the 16S rRNA gene is approximately 1500 bps long, meaning only a fraction of the entire gene can be sequenced. Conversely, long-read sequencing (e.g., Pacific Biosciences or PacBio) allows sequencing of the full-length 16S rRNA gene, enabling classification at the species level [15]. However, since there are differences in 16S rRNA gene databases, it becomes challenging to compare studies and assign sequences to lower taxonomic profiles. For instance, PacBio human fecal microbiota sequences were assigned more reads with the SILVA than the Greengenes database [16]. The study also found fewer sequences assigned to PacBio compared to Illumina MiSeq and whole-metagenome shotgun sequencing at the genus level [16].

We hypothesized that PacBio (full-length) sequencing can detect species-level changes in the equine microbiota induced by antimicrobial administration. The main objective was to characterize the gut microbiota of horses at the species-level pre- and post-TMS administration. A secondary objective was to compare Illumina with PacBio sequences. Full-length sequencing of the 16S rRNA gene showed significant decreases in alpha and beta diversity after antibiotic administration. All taxa with a relative abundance greater than 1% remained unclassified at the species level, indicating that the bacterial species present in the gut of horses remain largely unknown or that the current databases are not sufficiently comprehensive. Full-length sequencing showed greater changes in beta diversity than short-read sequencing.

## 2. Materials and Methods

### 2.1. Study Design and Animals

Nine adult horses of the asthmatic horse research herd, Faculté de médecine vétérinaire, Université de Montréal were enrolled. All horses were in remission and had no history of gastrointestinal diseases or antimicrobial administration during the previous 3 months. Horses received methylprednisolone 3 months prior to the study and an anti-helminthic (Eqvalan) and were vaccinated (Vetera Gold, Boehringer Ingelheim Animal Health Canada Inc, Burlington, VT, Canada) 2 months prior to the study. The animals were housed on turnout with shelter, kept on grass pasture, fed silage, and had access to a salt block.

All horses received trimethoprim sulfadiazine (TMS, 30 mg/kg) for 5 consecutive days per os twice a day. Horses were monitored daily (physical exam). Fecal samples were collected directly from the rectum before (PRE) and after (POST) antibiotic administration. Fecal samples were stored on ice after collection and then placed at −80 °C (thus, within 2 h of collection) until DNA extraction.

### 2.2. Microbiota Analysis for PacBio and Illumina Sequencing

Total DNA was extracted from the fecal samples using a commercial kit (DNeasy PowerSoil Kit, QIAGEN, Toronto, ON, Canada) following the manufacturer’s instructions. Full-length 16S rRNA gene was amplified by PCR using universal primers 27F (AGRRTTYGATYHTDGYTYAG) and 1492R (TASVGHTACCTTGTTACCGACTT) [17]. Amplicon libraries were created using a SMRTbell express template prep kit 2.0 (Pacific Biosciences) and sequenced using a Sequel 2 platform at the Delaware Biotechnology Institute Sequencing and Genotyping Center of the University of Delaware, Newark, DE, USA.

For Illumina sequencing, PCR amplification of the V4 region of the 16S rRNA gene was performed using primers 515F (GTGCCAGCMGCCGCGGTAA) and 806R (GGACTACHVGGGTWTCTAAT) [18]. Sequencing was performed using an Illumina MiSeq platform for 250 cycles from each end at the Génome Québec Innovation Centre.

### 2.3. Sequence Analysis

For the data acquired from PacBio sequencing, the software DADA2 [19] was used to exclude reads with poor quality and without the correct length (~1500 bp). Reads were grouped into amplicon sequence variants (ASVs), and the software SBanalyzer 2.4 (Shoreline Biome) was subsequently used for taxonomic classification using the Athena databank as reference [17]. Illumina data were evaluated by an established protocol using the software Mothur (Ann Arbor, MI, USA) [20], following a standard operating procedure previously recommended [21]. Reads containing more than 300 bps were excluded, and good-quality reads were aligned to the SILVA reference alignment. Chimeras were removed with the vsearch algorithm. Reads were classified according to the Ribosomal Database Project (2016 release), and reads belonging to the same genus (94% similarity) were clustered.

Richness was calculated by the total number of observed OTUs and by the Chao index, and diversity was estimated by the Simpson’s and Shannon indices. Subsampling using the smallest number of reads obtained in a sample was used to standardize non-uniform samples in an attempt to avoid introducing bias into the analysis.

### 2.4. Statistical Analysis

Bar charts were generated to visualize the most abundant phyla and species (>1%). Relative abundances were compared between sequencing technologies using paired Student t-tests. The linear discriminant analysis (LDA) effect size (LEfSe), which uses a non-parametric factorial Kruskal–Wallis with a subsequent Unpaired Wilcoxon test, was used to detect significant overrepresentation of taxa [22] PRE and POST treatment with TMS. An LDA higher than 3 and a *p*-value lower than 0.05 were considered significant.

Alpha diversity (number of observed genera, Chao, Simpson’s, and Shannon indices) was compared between samples PRE and POST antimicrobial administration by a t-test with two dependent means. The Kolmogorov–Smirnov test was used to assure normality, and the agreement between PacBio and Illumina results on each alpha diversity index was assessed using the Pearson correlation coefficient, considering a *p*-value < 0.05 as significant.

## 3. Results

### 3.1. Horses

All 9 horses remained clinically normal with no changes in temperature, behavior, appetite, respiratory and cardiac frequency, and gastrointestinal motility.

### 3.2. Characterization of the Horse Fecal Microbiota Using Long-Read Sequencing

The microbiota analysis before antibiotic administration revealed that, at the phylum level, Bacteroidetes were the most abundant (49.5%), followed by Firmicutes (24.1%), Fibrobacteres (17.5%), Spirochaetes (4.4%), and Cyanobacteria (1.2%).

The full-length analysis of the 16S rRNA gene failed to classify the dominant bacteria at the species level. Figure 1A shows that the most abundant taxa comprising the microbiota of the studied horses were unclassified Fibrobacter (16.6%), an unclassified, unknown species (13.2%) belonging to the Bacteroidales order, unclassified Tannerellaceae (8%), unclassified Clostridiales (7.8%), unclassified Bacteroidales (6.6%), unclassified Muribaculaceae (4.6%), unclassified Lachnospiraceae (4.1%), unclassified Treponema (4.0%), and unclassified Prevotella (2.0%).

### 3.3. Long-Read Microbiota Analysis after TMS Administration

Changes in relative abundance pre- and post-antibiotic administration are shown in Figure 1A. A significant decrease in relative abundance post-antibiotic administration was observed in unclassified Treponema (Figure 1B, *p* < 0.05) and unclassified Cyanobacteria (Figure 1C, *p* < 0.05). Conversely, unclassified Bacteroidales significantly increased post-antibiotic treatment (Figure 1D, *p* < 0.05).

The LEfSe analysis revealed 15 taxa significantly overrepresented in the microbiota of horses pre-antibiotic administration, all of which were unclassified at the species level, but no specific taxa could be associated with a sample collected post-antibiotic administration (Figure 2).

### 3.4. Comparison between Two Sequencing Protocols (Illumina Short-Read and PacBio Long-Read)

The relative abundance of the main phyla (>1%) found by PacBio and Illumina sequencing is presented in Figure 3.

Table 1 summarizes the results of alpha diversity indices obtained before and after TMS administration with both Illumina and PacBio. The correlation between results obtained with the two technologies was weak and non-significant to all the parameters evaluated (Table 1). Overall, PacBio sequencing was able to detect significantly more observed taxa (average = 60.95 and 196.67 for Illumina and PacBio, respectively, *p* < 0.0001). Results of the impact of TMS on alpha diversity indices (PRE versus POST) observed in the microbiota of horses using the two sequencing technologies are presented in Table 2.

Compared to the full-length sequencing data (Figure 2), the LEfSe analysis using data obtained from the Illumina short reads sequencing showed overrepresentation of *Phascolarctobacterium*, unclassified Subdivision 5, and unclassified bacteria at baseline (pre-antibiotic administration), and overrepresentation of Paraprevotella after TMS administration (Figure 4).

## 4. Discussion

The equine bacterial microbiota at the species level using long-read sequencing (PacBio) of the 16S rRNA gene was compared. However, all the most abundant reads failed to be classified at the species level. Noteworthy, samples from other environments sequenced in the same run and processed with the same bioinformatics protocol could be accurately classified as species. That suggests that either the bacteria present in the equine gut remain to be discovered or that the databases available for taxonomic classification do not contain the information required for the analysis of horse gastrointestinal samples. Other studies using PacBio to sequence the human fecal microbiota [23] and raw milk from mares [24] were able to obtain classification at the species level. This indicates that the bacteria present in the feces of the studied horses were not present in the database used for analysis (Athena). Even after classifying reads with the Ribosomal Database Project (RDP), results remained unclassified at the species level, indicating that the current databases need to be updated to better reflect the species present in the equine intestinal microbiota.

New methods of massive, untargeted bacterial culture (culturomics) have been used to isolate most of the bacterial species inhabiting the intestinal environment of the human gut [25]. Therefore, future studies using culturomics to determine the species present in the equine intestine could improve current databases. In addition, the development of horse-specific databases could also be developed for this purpose.

One advantage of full-length sequencing over short-read sequencing is the lack of primer bias. In short-read sequencing, primers target one or two of the 9 conserved regions of the 16S rRNA gene, while full-length sequencing targets the entire gene. For example, the V4 region that is the most used for microbiota studies has approximately 180 bps in comparison with the 1500 bps of the whole gene. Despite the fact that the bacteria could not be classified at the species level in the present study, full-length sequencing is more reliable than other technologies to classify bacteria at the genus level. This study confirmed that some bias may occur in short reads sequencing (e.g., more unclassified reads and fewer reads classified as Firmicutes).

PacBio was also able to detect higher richness and less unclassified bacteria compared to Illumina sequencing, and the agreement between the two technologies was weak, indicating that long-read sequencing might be more accurate for the study of alpha diversity. Despite the advantages of long-read sequencing, the higher costs and the incapacity to classify reads at the species level may negatively affect the feasibility of using those technologies in large equine cohorts.

Firmicutes have been consistently observed as the most abundant phylum in the horse gut, but there has been no consensus regarding the second most abundant phylum [13]. Bacteroidetes and Verrucomicrobia have both been considered highly abundant in horses, but methodological bias related to PCR amplification is most likely responsible for such differences [18]. Results of this study showed Bacteroidetes, Firmicutes, and Fibrobacteres as the most abundant phyla in equine feces. Interestingly, samples collected from the same horses at a different timepoint using Illumina sequencing of the V4 region confirmed Bacteroidetes as the most abundant phylum in that herd [26]. In addition, Fibrobacteres were overestimated by Illumina sequencing with regards to Firmicutes in this study. Noteworthy, the horses were part of a herd of asthmatic animals that have been shown to have similar fecal microbiota to healthy animals during remission.

Another important finding was the presence of a highly abundant unknown species of the Bacteroidales order. At this point, it is not clear if this is a true unknown species or the inability of the databanks to correctly classify this organism. This information cannot be extrapolated to the general equine population as it might be a particularity of this specific herd, but further investigations using selective culture-based methods are justified.

TMS administration was associated with significant decreases in richness but not diversity, indicating that this antibiotic mainly affects the less abundant bacteria. Several studies have investigated the impact of antimicrobials on the equine fecal microbiota [8,9,11,27,28,29], but such discussion is beyond the scope of this report.

The high inter-individual variability among the studied horses in response to TMS administration is intriguing. For instance, some horses reacted with an increase in Fibrobacteres, while others presented a marked decrease of this phylum. This finding highlights the concept of precision medicine currently applied in humans, which considers that each individual might respond differently to the metabolization of certain drugs and compounds, and it is reasonable to assume that the intestinal microbiota may also play an important role.

LEfSe analysis revealed overrepresentation of *Phascolarctobacterium* in horses before antibiotic administration, but the role in horses is unknown. The relative abundance of this genus was increased in humans with dysbiosis [30], however, and in children with autism [31]. Further, Subdivision 5, a genus part of the Verrucomicrobia phylum, was overrepresented in horses before antibiotic administration as well. This bacterium is considered part of the healthy equine gut repertoire [26,32]. Lastly, *Paraprevotella* was overrepresented in horses after antibiotic administration, but its role is unknown [33].

## 5. Conclusions

In conclusion, full-length PacBio sequencing of the 16S rRNA gene could not classify the equine intestinal microbiota at the species level but was able to detect greater richness and less unclassified bacteria compared to short-read Illumina sequencing. Bacteroidetes were the most abundant bacteria in feces, followed by Firmicutes and Fibrobacteres. For future studies, the current databanks should be improved by means of in-depth bacterial culture methods in order to classify the bacterial species present in the equine gut.

## Figures and Tables

**Figure 1 animals-11-02859-f001:**
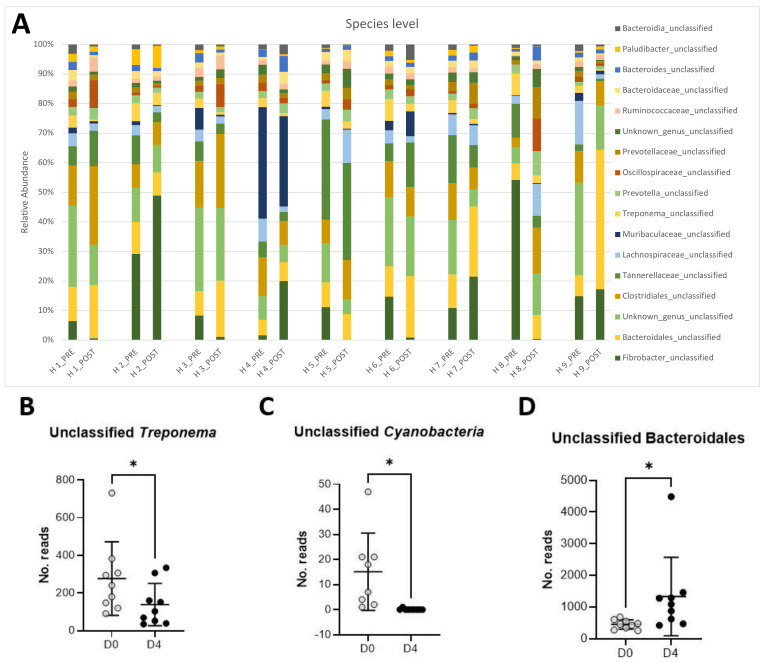
Changes in relative abundances of the main taxa (>1% abundance) present in the equine feces of 9 horses pre- and post-antibiotic administration of oral trimethoprim sulfadiazine (**A**). Relative abundance of unclassified *Treponema* (**B**), unclassified *Cyanobacteria* (**C**), and unclassified Bacteroidales (**D**) pre- (D0) and post-antibiotic administration (D4).

**Figure 2 animals-11-02859-f002:**
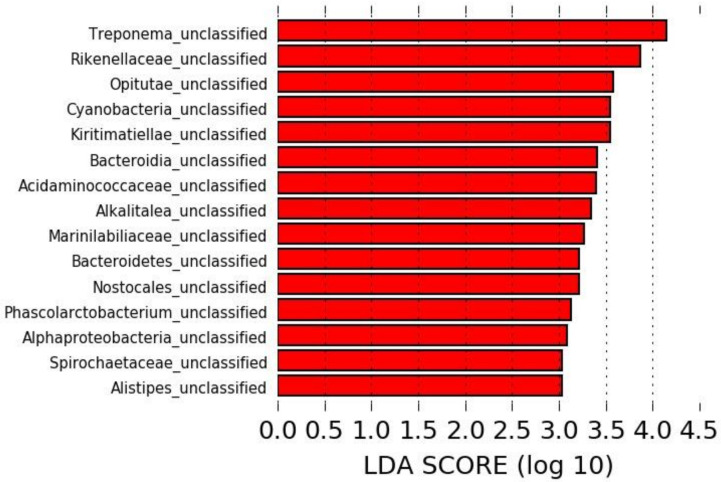
Linear discriminant analysis effect size (LEfSe) identifying taxa overrepresented in the microbiota of horses pre-antibiotic administration.

**Figure 3 animals-11-02859-f003:**
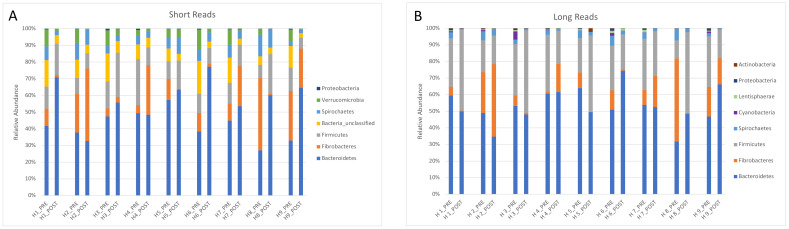
Relative abundance of the main phyla found in the feces of 9 horses investigated by Illumina short-read (**A**) and PacBio long-read sequencing (**B**).

**Figure 4 animals-11-02859-f004:**
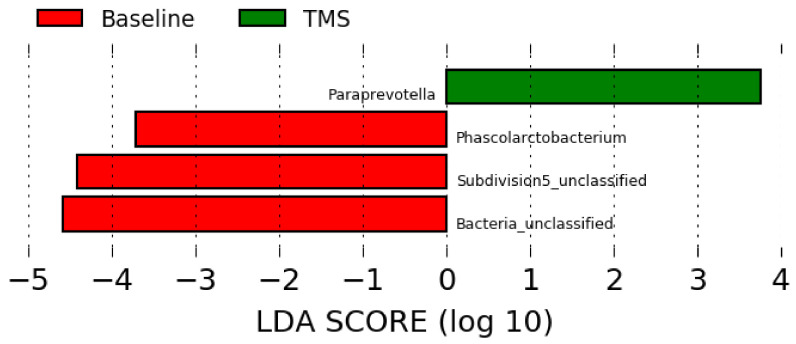
Linear discriminant analysis effect size (LEfSe) using data at the genus level obtained by short reads sequencing before (“Baseline”) and after (“TMS”) TMS administration.

**Table 1 animals-11-02859-t001:** Average and standard deviations (SD) of alpha diversity indices, total number of good-quality reads obtained, and correlation and *p* values comparing results of two different sequencing technologies (PacBio and Illumina) before and after treatment with oral TMS in horses.

	PRE TMS		POST TMS		
	PacBio	Illumina	PacBio	Illumina	Correlation (*p*-Value)
Good-quality reads	176,967	687,081	124,796	737,802	
Observed genera	224.22 (57.36)	68.43 (9.22)	169.11 (40.83)	53.40 (5.58)	0.4222 (0.081)
Chao	354.27 (86.95)	87.26 (14.74)	253.45 (51.57)	67.99 (5.29)	0.3576 (0.145)
Simpson’s	0.93 (0.08)	5.68 (0.98)	0.93 (0.04)	4.44 (1.93)	0.0724 (0.775)
Shannon	5.56 (0.99)	2.23 (0.17)	4.90 (0.73)	1.97 (0.37)	0.4519 (0.060)

**Table 2 animals-11-02859-t002:** *P* values obtained from a *t*-test comparing the impact of antimicrobial administration on alpha diversity indices evaluated by two different sequencing technologies (PacBio and Illumina).

	PacBio	Illumina
Observed genera	0.651	0.004 *
Chao	0.033 *	0.005 *
Simpson’s	1.000	0.197
Shannon	0.149	0.148

* values < 0.05.

## Data Availability

Data will be deposited at the SRA (NCBI) databank.

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
