# Peer review of "Species-Level Gut Microbiota Analysis after Antibiotic-Induced Dysbiosis in Horses"

_animals, 2021, doi:10.3390/ani11102859_

Round 1

Reviewer 1 Report

In the submitted paper the authors have described the results of the effect of antibiotic use on the faecal microbiota using long read sequencing and how this compared to the more frequently used short read sequencing. The paper is well written and presented. The topic is of interested to readers due to the long read sequencing becoming more common in bacterial DNA sequencing. I support the publication of this paper the journal Animals after the below minor comments are addressed. I have two comments on the presentation of the figures:

  • There are no y axis labels in the graph in Figure 1A and both graphs in Figure 3.
  • Could the colours that represent the bacterial phyla in figure 3 be changed to be the same for the two graphs? They are for Firmicutes, Bacteroidetes and Proteobacteria but it would be good if the other phyla were too so it is easier to compare the two graphs.

On line 187 you describe the differences in the bacterial groups identified by the two types of sequencing using LEfSe analysis. Could this be represented graphically? The authors have described the difference in bacterial diversity measures given by the two groups however there is no analysis reported that explores the different bacterial groups (or OTUs) identified by the two approach. Could such an analysis using LEfSe be incorporated into this paper? This could either be done comparing two groups: long read v short read sequencing. Alternatively, four groups (two types of sequencing and two groups of horses) could be compared at the same time.

It is interesting to see that reference database bias to human samples is also an issue with the long read sequencing of horse faecal microbiota. Are there any suggestions by the authors on how the the sequencing reference databases could be improved that could be added to the end of the conclusions?

Author Response

Thank you very much for taking the time to review our manuscript and for your valuable comments that have undoubtedly improved the quality of our work. The manuscript has also been revised by an English native speaker.

Reviewer 1:

In the submitted paper the authors have described the results of the effect of antibiotic use on the faecal microbiota using long read sequencing and how this compared to the more frequently used short read sequencing. The paper is well written and presented. The topic is of interested to readers due to the long read sequencing becoming more common in bacterial DNA sequencing. I support the publication of this paper the journal Animals after the below minor comments are addressed. I have two comments on the presentation of the figures:

  • There are no y axis labels in the graph in Figure 1A and both graphs in Figure 3.

AUTHORS: Thank you for pointing that out. The label “Relative Abundance” has been added.

  • Could the colours that represent the bacterial phyla in figure 3 be changed to be the same for the two graphs? They are for Firmicutes, Bacteroidetes and Proteobacteria but it would be good if the other phyla were too so it is easier to compare the two graphs.

AUTHORS: Great suggestion. The colours and the order in which they appear have been matched.

On line 187 you describe the differences in the bacterial groups identified by the two types of sequencing using LEfSe analysis. Could this be represented graphically?

AUTHORS: Those are the results obtained by Illumina sequencing. We have clarified that in the text and included the figure (Figure 4).

The authors have described the difference in bacterial diversity measures given by the two groups however there is no analysis reported that explores the different bacterial groups (or OTUs) identified by the two approach. Could such an analysis using LEfSe be incorporated into this paper? This could either be done comparing two groups: long read v short read sequencing. Alternatively, four groups (two types of sequencing and two groups of horses) could be compared at the same time.

AUTHORS: That is exactly what we did with the LEfSe, but we analyzed the data independently (Figure 3 and 4). We believe that using four groups at the same time would not be appropriate because data were analysed using different software and pipelines.

It is interesting to see that reference database bias to human samples is also an issue with the long read sequencing of horse faecal microbiota. Are there any suggestions by the authors on how the the sequencing reference databases could be improved that could be added to the end of the conclusions?

AUTHORS: We have stated in the discussion that culturomics studies could be used to discover new species present in the equine gut and that the development of horse-specific databases could also be used. We have included this also in the discussion, as suggested.

Reviewer 2 Report

I appreciate all the work that went into this study. The topic is highly interesting, however, the authors can need to clarify some aspects.

Materials and Methods: horses included in the study were asthmatic. In a preview research from the same team using the same horses the results showed that asthmatic horses had different response in the fecal microbiota than healthy horses. I think this is a important point that need to be clarify. If the response is different is (asthma is a chronic disease) should be included in the tittle and in the discussion, because maybe this results can not be extrapolated to healthy horses.

Otherwise, the article is very interesting and the quality is high. 

Author Response

I appreciate all the work that went into this study. The topic is highly interesting, however, the authors can need to clarify some aspects.

Materials and Methods: horses included in the study were asthmatic. In a preview research from the same team using the same horses the results showed that asthmatic horses had different response in the fecal microbiota than healthy horses. I think this is a important point that need to be clarify. If the response is different is (asthma is a chronic disease) should be included in the tittle and in the discussion, because maybe this results can not be extrapolated to healthy horses.

Otherwise, the article is very interesting and the quality is high.

AUTHORS: Thank you for your comments. In the previous study those horses responded differently to exposure to moldy hay, but had similar microbiota to healthy horses during remission. As described in the material and methods, all horses were in remission for the present study. We have acknowledged that in the discussion.

Reviewer 3 Report

The authors tried to charectarize TMS induced dysbiosis in horses by using geneome sequencing techinques. The methodology is sound and the results are described adequately. However English language and the discussion are not well written. May be the authors did not expect such results. As the topic is interesting and might have broader applications to the scuientific community, I would recommend to revise the discussion with rigorous emphasis on the reasons why such results may have came out, the meaning of these results and the prospects of these results in the future scientific research related to this topic. 

Author Response

The authors tried to charectarize TMS induced dysbiosis in horses by using geneome sequencing techinques. The methodology is sound and the results are described adequately. However English language and the discussion are not well written. May be the authors did not expect such results. As the topic is interesting and might have broader applications to the scuientific community, I would recommend to revise the discussion with rigorous emphasis on the reasons why such results may have came out, the meaning of these results and the prospects of these results in the future scientific research related to this topic. 

AUTHORS: Thank you very much for your comments. We have expanded the discussion and the manuscript has been revised by a native English speaker.